# Investigation of Çankiri City Railway in Scope of Greenway

Umut Pekin Timur *, Ferhat Özden, Pakize Ece Erzin and Özgür Burhan Timur

Landscape Architecture Department, Faculty of Forestry, Cankiri Karatekin University, Cankiri 18100, Turkey; frht.ozden@gmail.com (F.Ö.); ecerzin@gmail.com (P.E.E.); ozgurtim@hotmail.com (Ö.B.T.)
* Correspondence: umutpt@karatekin.edu.tr; Tel.: +90-376-2122757

**Abstract:** Nowadays, the importance of greenway applications in planning studies has increased to meet the increasing environmental problems in cities, the loss of open spaces, and the increasing need for recreation. The purpose of this study is to evaluate within the scope of the greenway about the part of 11 km of the Çankırı railway passing through the city center by considering the planning and design dimension. The research was carried out in four stages: literature review, analysis of natural and cultural landscape characteristics, preparation of a conceptual plan, and design suggestions. The area has a straight aspect, the slope is 0–3%, and it is an alluvial structure. It was determined that there are registered historical buildings, but structural areas are intense. According to plan, a green area and a bicycle path arrangement was proposed at the side of the railway. Also, along the existing road routes, a walking or bicycle path was proposed to provide connection between open green areas in the city. In the design process, appropriate landscape design solutions were developed for the station area, and it was proposed that street area and some intersection points be pedestrianized.

**Keywords:** rail with trail; greenway; railway; urban sustainability; open and green areas; recreational greenway; GIS; CSS; Çankırı

## 1. Introduction

The urban areas inhabited by the majority of the world population are most appealing as settlement areas which offer various opportunities for urban people. But today they also face many environmental problems due to rapid urbanization, population growth, and developments in industrialization. In urban issues, not only those of an environmental dimension stand out but also those at economic and social levels [1]. According to estimates made by the United Nations and many sources, it has been clearly stated that this tendency will continue throughout this century, and by 2050 70% of the world population will be living in cities [2]. Parallel to this, the existing problems in cities will certainly increase. The solution will be possible with the implementation of development plans including sustainable and holistic approaches.

With environmental, economic, and social benefits, greenways play an important role solving problems that originate in urbanization [3].

Greenways are linear open spaces such as canals and scenic roads that are set along the riversides, hillsides, or valleys and converted to a recreational use along the railways. These linear open spaces connect parks, natural reserves, and cultural or historical sites to each other and settlements [4].

While public parks are a static concept, the greenways are a concept that allows people and wildlife to move in terms of the functioning natural processes by creating a green transportation network [5].

Greenways are classified into five grades in terms of their basic characteristics [6]. These features are listed below:

- Linearity: Their spatial forms are linear, with this feature contributing to recreation and ecology.

- Linkage: They make connections between life spaces by forming associations with landscape forms of all scales with their binding qualities.
- Multi-functionality: They can be multifunctional, with ecological, recreational, and cultural functions.
- Consistency with sustainability: They are consistent with sustainable development because they are not created only for protection of nature and environment, but also for people's use.
- Contribution to landscape planning: They present different spatial strategies depending on the benefits and features of networked linear open spaces.

These features offer a variety of environmental, economic, and social benefits to urban areas. Environmental benefits can be summarized as conserving and repairing natural areas, protecting and supporting biodiversity, protecting water resources, reducing flood hazards, and providing a living space for plants and animals [7]. Economic benefits: to reduce public investment and private spending, to create an affordable strategy for securing outdoor recreation, to provide tourism attraction, and to increase property value in neighboring areas. Social benefits: to provide protection of historical and cultural resources, to create recreation and alternative transportation facilities, to strengthen social ties, and to increase the quality of life and environmental awareness [8].

Especially with the decline of open space and the need for recreation in the major cities of the US in 1980s, attention has been drawn to the greenways dating back to 1865 [4,9].

Although there are different greenway categories made by different scholars, the most common classification made by [4] into five categories. These are:

- Urban riverside greenways
- Recreational greenways
- Ecologically significant natural corridors
- Scenic and historic routes
- Comprehensive greenway systems or networks

While a significant portion of the greenways are intended for recreation or nature conservation, some of them are planned to cover both purposes. In this context, it is possible to find examples of active or inactive railway routes that are used for recreational greenways in many countries.

Especially after the 1920's in America with fall of railway use, some routes have been retired. With the increase of the urban population in the country in America, people tried to escape urban pressures through bicycling, horseback riding, and hiking. For this purpose, several old railroad beds have been converted to trails. Congress enacted the Trails Act in 1968 to establish a nationwide system of nature trails. In 1983, railway corridors that were out of use for this period were also included. Approximately 141,000 miles are now in use, but it is predicted that another 3000 miles will be abandoned every year through the end of this century [10].

One of the important steps for the use of railway routes as a greenway belongs to the Rails-to-Trails Conservancy (RTC) in America. RTC is a non-profit organization dedicated to building a nationwide greenway network from old railway lines and connection corridors to create healthy areas for people. The RTC provides a range of data and demonstration projects to help raise awareness of railway trails and greenways and improve local and state policies and practices that support railway trails. It has helped develop over 31,000 km of railway trails throughout the country and has provided technical support for the construction of potential railways within the greenway for thousands of kilometers [11].

According to the European Greenway Association, the transition from the railway to the greenway in Europe has similarities with the US, and it is usually created for a "safe, continuous and environment-friendly alternative transportation route". It is possible to find many examples of railway to greenways in America and Europe. For example, in Germany there are 550 greenways [12].

A beautiful and well-known example is the High Line in Manhattan, USA, which has recently been turned off and turned into a greenway. The High Line, which went out of use

in the 1980s, is a railway line that was moved up the road in the 1930s during "the West Side Improvement Project". In 2006 it was converted into a linear park of about 2.3 km [13].

The RTC organization supports the use of greenways (rail with trail) in active railway corridors. This usage is safe, widespread, and increasing in number day by day. Clarion-Little Toby Creek Trail, Lehigh Gorge State Park Trail, Richmond Greenway, Frisco Trail, and Charlotte Trolley Trail are examples of this usage.

An example of an active railway is the Richmond Greenway in California. The 18-mile rail-trail (2 mile rail with trail) meanders along the wild and scenic Clarion River and Little Toby Creek through Elk and Jefferson counties between the charming small towns of Ridgway and Brockway [14].

The railway route greenways, which are both converted from the unused and actively used railway to the greenway, serve many purposes. The research that RTC conducted on 88 active railway-route greenways in current use in 33 different states of the US revealed that all trails were open to pedestrian use while 95% permitted bicycle use and most other non-motorized vehicles (skates, skateboards) [11].

The Context Sensitive Solution (CSS) criteria should also be taken into consideration in planning studies because greenways provide an alternative transport corridor for urban people. CSS, which connects the lands and settlements where streets, roads, and highways pass, is a theoretical and practical interdisciplinary approach that, by paying attention to the beauty of the landscape and protecting natural, historical, and aesthetic areas, ensures environmental sustainability [15,16].

In this context, the railway line in active use passing through the city center of Çankırı is seen as valuable by virtue of its linearity and its central position. The aim of this study is to plan and design about 11 km part of Çankırı railway passing through the city center within the scope of greenway. In the conceptual plan created in line with the principles of greenway planning, recreational greenways that will allow pedestrian and bicycle use were proposed for the city, considering almost no transportation options other than cars and public transportation, and almost no areas for bicycle use. CSS criteria were taken into consideration for providing transportation connections in the plan. According to this, a green area and a bicycle path arrangement was proposed at the side of the railway. Also, in the existing road routes, walking or bicycle paths were proposed to provide connections between open green areas in the city. In the design process, appropriate landscape design solutions were developed for the station area, also street areas and some intersection points were proposed to be pedestrianized.

## 2. Materials and Methods

The main material of the study is the 11,691 m railway line periphery extending between Yaren Gate and Kastamonu Road-New Road connection in Çankırı City and city center (Figure 1).

Çankırı city lies between 40°30′ and 41° north latitudes and 32°30′ and 34° east longitudes in the northwestern part of Central Anatolia. Kastamonu and Zonguldak lie to the north of the city, Bolu to the west, Ankara to the south, and Çorum to the east; the surface area is 2210 km$^2$ and the sea level is 736 m [17]. The city center consists of 24 districts.

Today, there is no passenger transportation on the Çankırı railway. Due to heavy use of highway in 2012, passenger transportation was closed. Only freight transport is carried out.

In order to obtain data in the research and to interpret and evaluate the data obtained, various scales, maps (the contour lines, soil, hydrology, development plan, transportation, etc.), and reports were used. Geographic Information Systems (ArcGIS 10.3), design program (Autocad 2017), graphic design program (Photoshop CC 2017), and 3D modeling software (SketchUp 2017), etc. were used at various stages in the study.

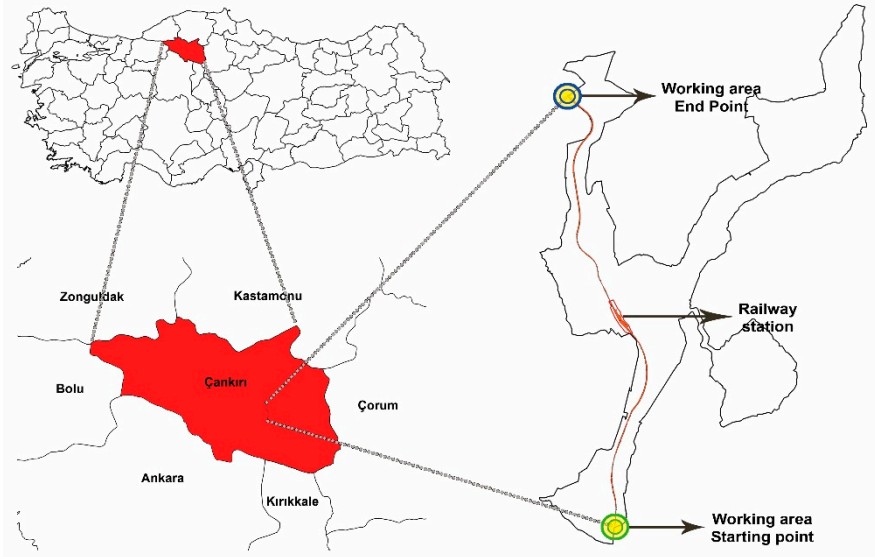

**Figure 1.** Location of the research area (Original, 2017).

Also, oral interviews with the related institutions and photographs taken in the area were used as auxiliary materials.

The purpose of the greenway in greenway planning process is the most important factor in determining the method. Greenways can be designed to serve one purpose or more than one purpose. Because of these reasons, the greenway planning process is varied. But generally it consists of three stages: inventory-analysis, preparing a conceptual or draft plan, and the preparation of the master plan [18].

In the study, creating the greenway was planned and designed according to [3,18–23]. The method was carried out in 4 interrelated stages (Figure 2).

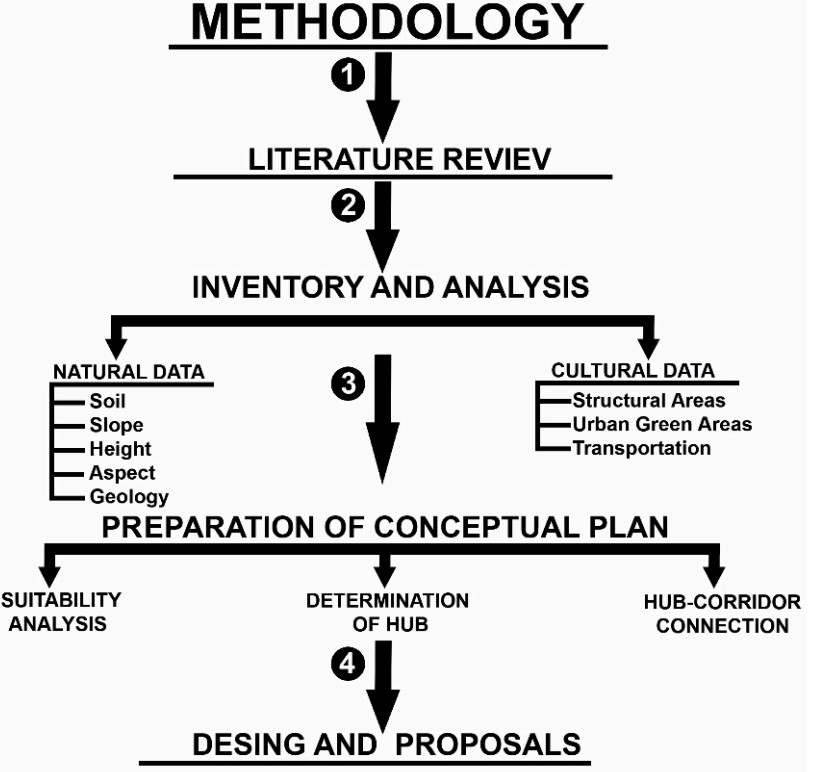

**Figure 2.** Methodology of the research (Original, 2017).

In the first stage, the literature was searched for the purpose of this study and necessary data were collected. In addition, the boundaries of the research corridor were determined. Although the width and length of the corridor in greenway works vary according to the purpose, it is recommended that it be as wide as possible [18,24].

Subsequently, the obtained and produced data with GIS were analyzed. As a result of the analysis made at the third stage, a proposed conceptual plan was prepared. While preparing this plan, first the suitability analysis of [25,26] was made on the maps created to determine the areas suitable for greenway use. Then, to create an effective greenway plan, "hub and spoke model" used by [21,27] was targeted. The hub areas (especially open green areas and recreational areas) determined from the development plan's map were evaluated together in terms of providing connection with potentially suitable areas. If those suitable areas cannot link the defined hubs, the hubs will be linked to the trails that contain various uses (bicycle and pedestrian ways) inside them. Also, considering the contribution of greenways to urban transportation, CSS was also taken into consideration while preparing the plan.

At the last stage, in the direction of the conceptual plan, the station area, an intersection, and an area for pedestrianization on the route were designed and have been made suggestions about the field.

## 3. Results and Discussion

A total of 1500 m wide corridor was determined, 750 m from both sides of the railway, and this corridor mainly was analyzed. These analyses were evaluated under two headings according to natural and cultural data using GIS.

### 3.1. Natural Landcape Characteristics

- Height groups: The height range varies between 650 m to 850 m. These values are suitable for human comfort.
- Slope groups: Slope in greenway studies is a very important criterion in planning activities. In the map, the slope groups were taken into account under seven categories with percentage (%) slope type. The groups were classified as suitable for greenway studies [28]. According to analysis results, most of the area has a slope of nearly 0–3%. In addition, it was found in areas with 3–8% slight slope and 8–12% slope. While 5% slope is suitable for multi-purpose greenways, 3% max is preferable. 8% slope is suitable for bicycle use. Therefore the area for greenway use mostly has a suitable slope.
- Aspect status: Aspect is defined as the direction of the slope surface. It is measured by 0–360° clockwise starting from the north. Since the areas with a straight slope do not point in any direction, −1 value was assigned. According to Akar et al. (2006), the aspect factor plays a guiding role in the natural environmental characteristics such as humidity, precipitation, wind, and sunshine duration and intensity in the areas [3]. Northern aspects are cooler and shady; southern aspects are hot and sunny [29]. Most of the corridor has a flat area. Some areas and the parts where the train station is located have the east and northeast aspect. This situation will not constitute an obstacle for the corridor with an appropriate vegetative arrangement. If the evergreens are positioned on the north of the trail, they protect the trail from cold winter winds. The deciduous trees should be positioned on the south of the trail. These trees provide shadow during summer and they make the trail benefit from sun rays by defoliating during winter [28].
- Large Soil Groups: There is brown soil and alluvial soil groups throughout the corridor. The amount of alluvial soil is about 300 hectares. Alluvial soils are deep, permeable, easily processed, fertile soils. There is no soil data in the built areas. The station area is also located in the area where there is no soil data.
- Land use capability classes: There is no soil data in the built areas. The station area is also located in the area where there is no soil data. But, most of the corridor has Class

I agricultural land, as well as class IV, VI, and VIII areas. It should be ensured that 1st class lands are not used for non-agricultural purposes.

- Risk of erosion: Most of the area has very slight erosion risk. Erosion risk situation was created by overlaying slope, vegetation density and soil data.
- Visibility analysis: This analysis was carried out along the railway line. A large part of the environment can be seen from the area. The building heights were not taken into account in the analysis; the field of view may be interrupted due to floor heights and construction.
- Hydrology: Stream lines in the study area were analyzed. Only Tatlıçay passes through the area. In a certain part of the study area, the stream flows parallel to the railway line.
- Geological structure: Alluvial rock structure is visible in the study area. Built areas are located on this structure. These areas are important for agriculture and are not suitable for settlement in terms of flood and seismicity. Considering that Çankırı province is in the second degree earthquake zone, this constitutes a significant risk.

### 3.2. Cultural Landcape Characteristics

- Structural areas: Structural areas were examined through the city development plan and in this context, conditions such as dwelling density, building use cases, and registered buildings were analyzed. Accordingly, it was determined that there is medium and low density in the northern part, dense in the center and near the station area, and less intense in the southern part of the railway passing through the zoning boundaries.
- Urban green areas: Areas such as parks, reforestation sites, vineyards and gardens, forests, and central reserve areas have been identified green areas. Potential greenway routes that will ensure the connectivity between these areas were determined. According to the calculations made from the land use map, the amount of green areas per person in Çankırı city center is 5.7 square meters. According to the Spatial Plans Construction Regulations in Turkey, the amount of green areas required per person in urban areas should be at least 10 square meters [30]. The number of green areas per person in Çankırı city center is quite below the value it should be. At this point, the contributions of greenways to cities as linear open green areas are of great importance.
- Transportation: Çankırı province is located on the D-765 highway connecting Ankara and Kırıkkale to Kastamonu. The road and the rail transport networks were mapped. According to this, there are 3 and 4 degree roads in and around the study area. Transportation in the city is provided by public transport and private vehicles. Bicycle use is intertwined with existing traffic; it is not separated. This situation poses an important problem in terms of life safety. The railway line passing through the city is the Ankara—Zonguldak railway line. This line was used for passenger and freight transport in the past, but today it is only used for freight transport.

### 3.3. Planning and Design

3.3.1. Greenway Conceptual Plan

The natural and cultural data above overlap with each other and suggested greenway routes were determined. In this phase, slope, large soil groups, hydrology, structural areas, urban green areas, and transportation layers were used (Figure 3).

As [19,27,31–33] stated in their studies, the proposed greenways, which are suggested in this plan, will prevent building in areas with erosion risk, agricultural lands, and alluvium areas. Greenways to be implemented contribute to the protection of biological diversity and water resources. The environmental benefits of greenways vary according to the scale of the project [34–36]. It is also stated that the wider greenways provide a greater impact on the environment.

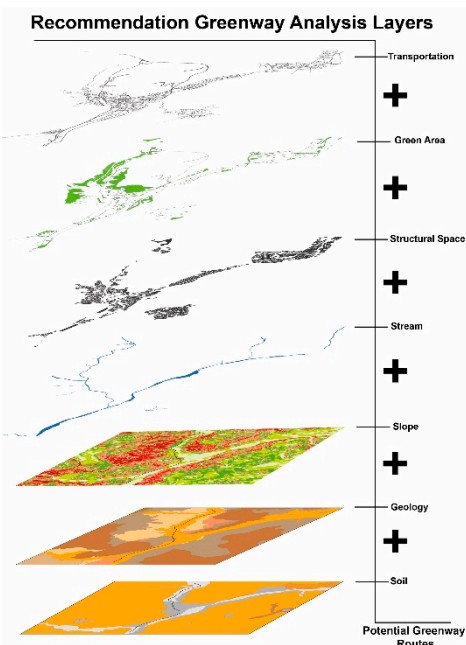

**Figure 3.** Used layer for Conceptual Greenway Plan (Original, 2017).

A large part of the planned route of the proposed greenways passes through the densely built urban area. Therefore, it is considered that the environmental benefit may be less than other benefits. However, the environmental benefits will increase over time, as the greenways themselves are not a source of pollution.

In the direction of the plan, a green zone of 2 m was left from the border of the railway line (parcel limit) for bicycle route arrangement on the side of the railway. From this distance, a bicycle route was proposed with 2.4 m. width as round-trip. In addition, it has been proposed to build bicycle or pedestrian paths based on existing road routes in order to connect with open-= green areas (hubs) in the city (Figure 4).

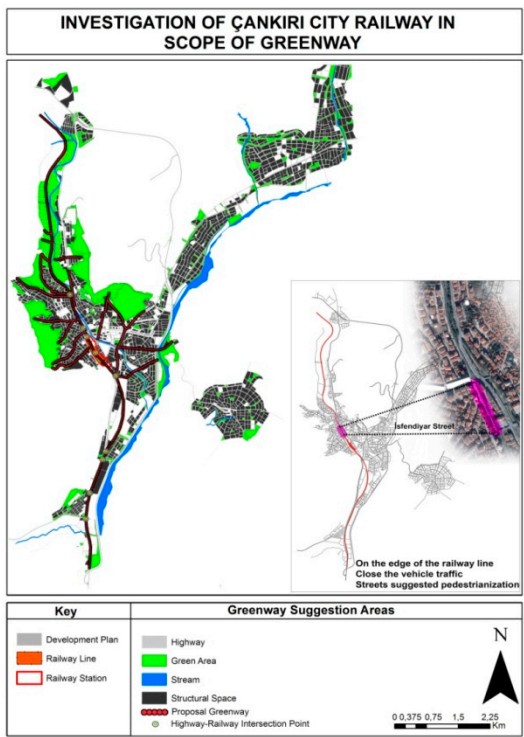

**Figure 4.** Conceptual Greenway Plan for Çankırı City (Original, 2017).

### 3.3.2. Recommendation Pedestrian Area for Isfendiyar Street

In the scope of the research, a pedestrianization for Isfendiyar Street with a width of 7 m was proposed for pilot purposes. 2 m of the street was reserved for building front-shares and part of the 5 m was proposed for arrangement. Figure 5 shows situation after project implementation of İsfendiyar Street.

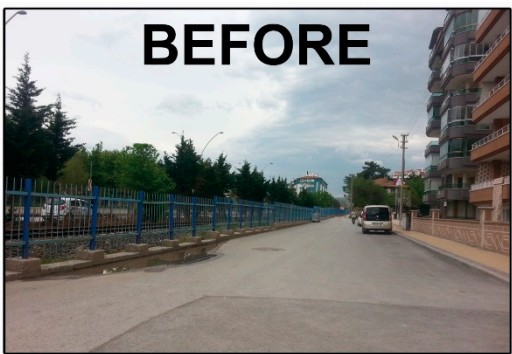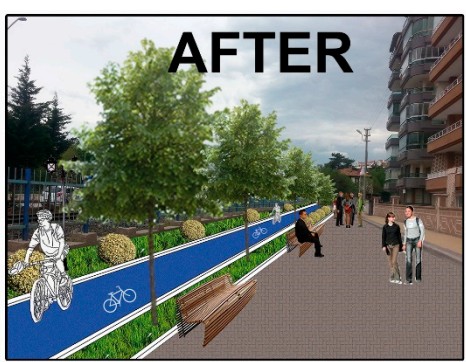

**Figure 5.** A view from İsfendiyar Street (before and after) (Original, 2017).

In this study, the basic features of linearity and connection formation, which are common points in the definitions of greenways in the literature [6,36] and in the planned study, were taken into consideration, and cycling and walking path opportunities were provided for the citizens. These trails created for recreation also serve the purpose of alternative transportation.

As Benedict and McMahon also Keith stated in their work concerning one of the important open green areas in the urban environment [37,38], greenways increase the amount of green space and air quality in the area. In a study by Novak (1994), the greenways and trails, especially with plants, avoided the $1 million spent on air pollution prevention in Chicago, Illinois, thanks to the air filtering properties of the plants [39]. With the afforestation around the proposed greenway, it has been thought to both increase the amount of green space in the city and increase the air quality of the city by providing an air flow corridor.

Isfendiyar Street section, where planning works were carried out, was planned as an area to be used to eliminate the lack of walking and gathering areas caused by urbanization. This planning contributes to the social sustainability dimension by strengthening social ties for neighboring residents and citizens of the city [7,40].

### 3.3.3. Recommendation for the Intersection Point

Level crossings at the intersection of the railroad and the highway are monitored by attendant and automatic signal systems (Figures 6 and 7). Our line has five uncontrolled, two bridges, and two controlled passes.

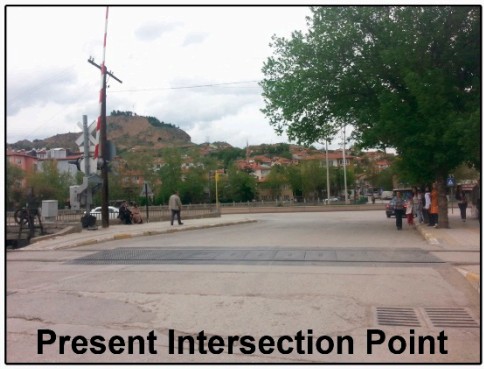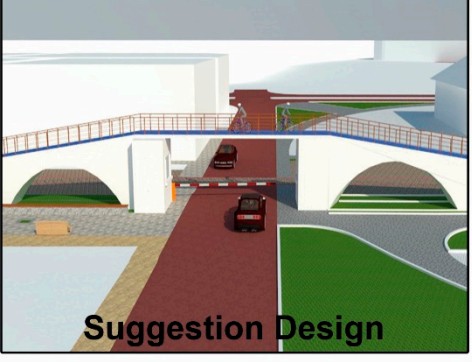

**Figure 6.** Proposal of intersection point (Original, 2017).

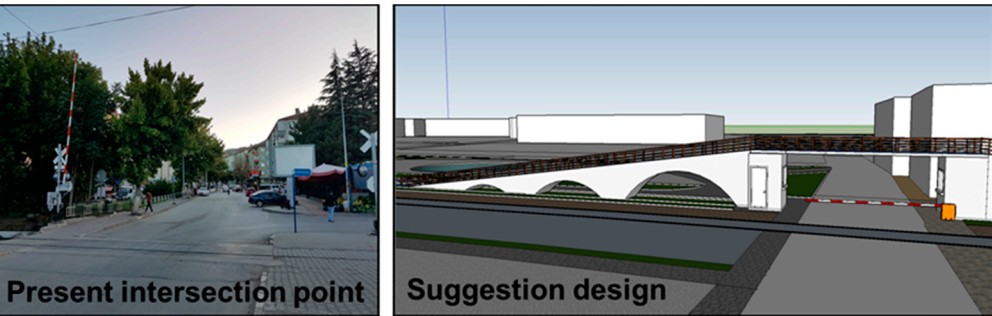

**Figure 7.** Proposal of intersection point (Original, 2017).

In the study, the intersection points planned by considering the CSS criteria [41] provide the connection between new alternative transportation routes and urban roads.

### 3.3.4. Station Area Landscape Design

The design area with seven historical buildings covers an area of approximately 16,000 m$^2$. These buildings have been preserved in the design (Figure 8).

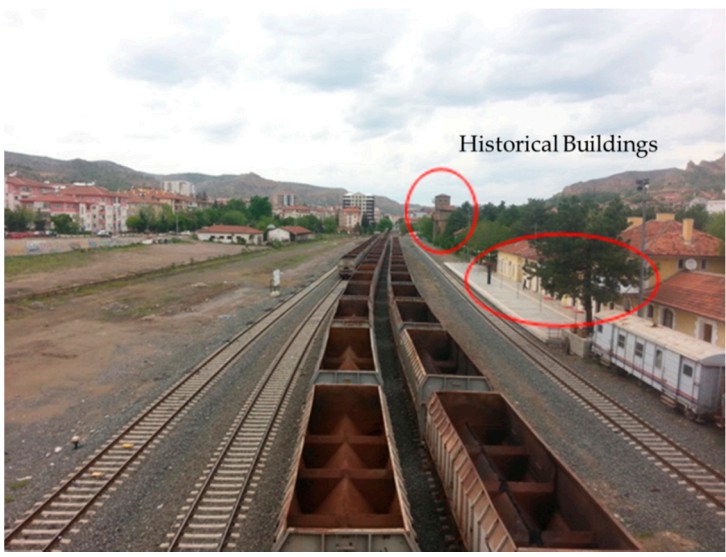

**Figure 8.** Historical buildings in station area (Original, 2017).

The design was made to support the suggestion of the bicycle path at the side of the railway line. For this reason, bicycle route and bicycle rental and sales places were also included in the area. Station areas were designed as squares and including arrangements such as eating and drinking areas, local sales area, water surfaces for scenery purposes, sitting rest areas, concert areas, and a railway open air museum (Figure 9).

Station area designed by taking into account the multifunctional basic feature of greenways provides the opportunity for recreation to city people while ensuring the protection of historical and cultural areas. Multifunctionality is a feature that often limits planners [6,19]. The plan and design made includes various functions such as alternative transportation, historical protection, and recreation.

The station area connects the recreation areas, walking and bicycle paths within the planned greenway with the urban transportation system. Also, as noted in the work of [21,36,42], greenways have a value that can increase the property value in neighboring lands.

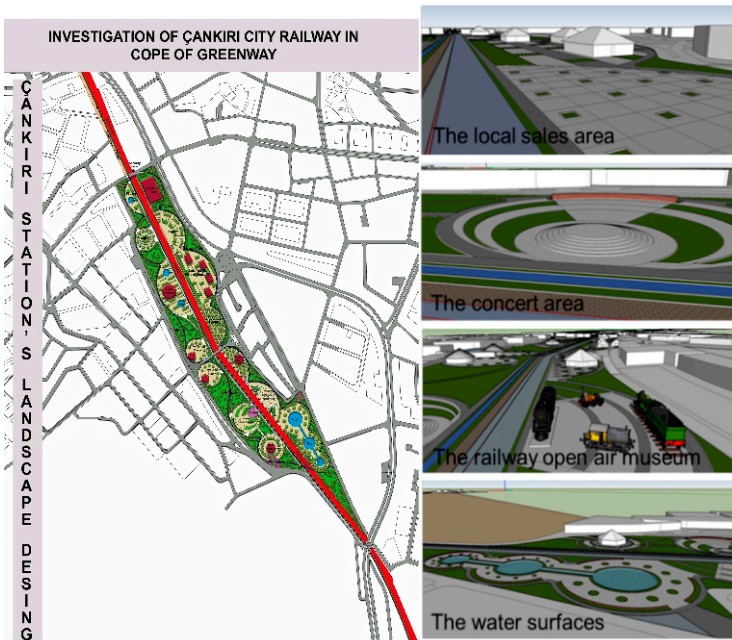

**Figure 9.** Landscape design for the Station Area (Original, 2017).

The main goal of the 'Habitat-2 City Summit' held in 1996, when sustainability was brought to the dimension of the city and its living environment, is to have healthy, safe, equal and sustainable cities, towns and villages now and in the future [43,44]. Development plan of Turkey [45] supports the expansion of pedestrian-bicycle transportation and increase of open green spaces for the scope of sustainability and livable environment. In this respect, the development plan coincides with the conceptual plan proposed in the study. There is no greenway practice in the country, especially for active railways. The conceptual plan is a study that can be applied by presenting it to the local administration.

Greenways can follow man-made corridors as well as natural corridors. This plan consists of 11,691 m of railway and greenways following the existing transportation axes. In future studies, a regional greenway system can be established by planning a comprehensive greenway system with the natural corridors to be added to the project and the continuation of the railway line.

## 4. Conclusions

Greenways are an effective tool in solving the problems arising from urbanization with the environmental, economic, and social benefits they provide, as well as a sustainable approach because it observes the protection-use balance in planning. For this reason, greenway planning has gained importance day by day and has become widespread in many countries.

In this study, a conceptual greenway plan was created by analyzing approximately 11 km of the railway line in active use and greenways following the existing transportation axes in Çankırı in line with the greenway planning principles. This conceptual plan was formed with an integrated approach that took the advantage of the opportunities and evaluated the present unities. The goals of the plan are listed below:

- Connections between the hubs that are present,
- Alternative transportation,
- Spaces for people,
- Recreational opportunities for the public.

According to this plan, proposal designs were created at the intersection points of the station area and highway and railway axes. In addition, the pedestrianization of Isfendiyar

Street and the arrangement of a bicycle path along the railway line were proposed. On the Tatlı Su Adası Street, only pedestrianization and bicycle path crossings were suggested.

Pedestrian and bicycle paths will be created on the existing road routes that slope allows in order to provide connection with the open green areas of the city.

Planning and design studies have been prepared to include minimum interruption for users. The project complies with the CSS criteria as it creates areas for safe use that preserve the landscape and the aesthetic, historical, and natural resource values of the region.

Planting trees and shrubs suitable for Çankırı conditions in the proposed trails will also be visually and functionally supportive.

With this plan will be offered the creation of both walking and cycling opportunities for the people of the city and physical activity; at the same time, air quality will be supported by increasing the amount of green space. As a result, the project prepared will contribute to effective transportation, a healthy society, and a healthy environment in the city.

The implementation of this plan and design should proceed in cooperation with different professional disciplines, and ongoing public participation should be ensured.

In the USA, where successful practices in greenway planning are exhibited, it is extremely important to ensure public participation in organizations. Public participation through organizations will increase the success from determining the goals and objectives of the greenway project to the implementation of the plan. For this purpose, an advisory committee should be formed to represent the public.

**Author Contributions:** All authors conceived and designed the article. Conceptualization, U.P.T. and P.E.E.; data collecting, F.Ö., P.E.E. and Ö.B.T.; land surveying and photographing, Ö.B.T. and F.Ö.; translating for the theoretical structure, P.E.E.; methodology, U.P.T., F.Ö., P.E.E. and Ö.B.T.; analysis, U.P.T. and F.Ö.; software, F.Ö.; visualition, F.Ö. and Ö.B.T.; writing, Ö.B.T.; revision, U.P.T. All authors have read and agreed to the published version of the manuscript.

**Funding:** This research received no external funding.

**Institutional Review Board Statement:** The study was conducted according to the guidelines of the Declaration of Helsinki. This study does not require live animal experiments or poll with humans.

**Informed Consent Statement:** Informed consent was obtained from all subjects.

**Data Availability Statement:** Not applicable.

**Conflicts of Interest:** The authors declare no conflict of interest.

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
