# Peer review of "Investigation of Çankiri City Railway in Scope of Greenway"

_sustainability, doi:10.3390/su13063055_

Round 1
Reviewer 1 Report
The manuscript mainly deal with Çankiri City Railway strructure within greenway properties, located in Turkey. It looks like well prepared original research and adress an important issue for landscape applications. However, some small modifications should be suggested for better presentation of manuscript. I recomended that the abstract of manuscript should be rewritten. In that section, only important findings should be presented rather literature informations. The Introduction aand Material and Method section should also be shortoned to make clear and easy of understanding that study.
Some pictures from historical buildings of study area could be added to text for better presentation.
I think that manuscript is well prepared and well suited with literature. It could be ACCEPTED after these small revisions.
Author Response
Dear Reviewer,
- Literature information removed from abstract
- CSS added to keywords
- Introduction shortened
- CSS related information added to introduction
- CSS information added for Material Method, the section was shortened
- 7 moved to page 11
- Figure 8 added on page 11
- The name of old figure 8 has been changed to figure 9 and moved to page 12
- 2 new articles numbered 15 and 16 added to references
- Next reference numbers have been changed
Regards

Reviewer 2 Report
In the introduction or literature review, author should talk about the context sensitive solution (CSS) for this study. You may find the CSS approach in planning urban transportation. Also relate your study with CSS approach in the methodology part.
Authors should improve the conclusion part showing the discussion and future study.
Author Response

(The authors gave the same response as above.)
